# Feature Transformers: A Unified Representation Learning Framework for Lifelong Learning

## Abstract

Despite the recent advances in representation learning, *lifelong learning* continues to be one of the most challenging and unconquered problems. *Catastrophic forgetting* and *data privacy* constitute two of the important challenges for a successful lifelong learner. Further, existing techniques are designed to handle only specific manifestations of lifelong learning, whereas a practical lifelong learner is expected to switch and adapt seamlessly to different scenarios. In this paper, we present a single, unified mathematical framework for handling the myriad variants of lifelong learning, while alleviating these two challenges. We utilize an external memory to store only the features representing past data and *learn richer and newer representations incrementally* through transformation neural networks - *feature transformers*. We define, simulate and demonstrate exemplary performance on a realistic lifelong experimental setting using the MNIST rotations dataset, paving the way for practical lifelong learners. To illustrate the applicability of our method in data sensitive domains like healthcare, we study the pneumothorax classification problem from X-ray images, achieving near gold standard performance. We also benchmark our approach with a number of state-of-the art methods on MNIST rotations and iCIFAR100 datasets demonstrating superior performance.

## 1    Introduction

Deep learning algorithms have achieved tremendous success on various challenging tasks like object detection, language translation, medical image segmentation, etc. Lifelong learning - the ability to adapt, benefit and sustain performance post deployment with more data and feedback, is an important goal for artificial intelligence (AI) and extremely crucial for sustained utility of these algorithms in domains like healthcare (learning from rare cases), self driving (learning new object detectors), etc. While there is no single standard definition of lifelong learning, most of the research in this field can be classified into one of the following sub-categories:

1. Incremental learning - encountering new data, with no change in distribution, for same task
2. Domain adaptation - data from modified target distributions but for the same task
3. New-task learning - data from tasks that were not presented before

Majority of the successful techniques study these variants in isolation. However, a realistic lifelong learning scenario would involve a mixture of these manifestations over time. Another major impediment to a successful lifelong learner is *data privacy*. In domains like healthcare, it is near impossible to have data access beyond scope (both time and geography) which leads to *catastophic forgetting* (McCloskey & Cohen, 1989)- the inability of machine learning models to retain past knowledge while learning from new data. In this paper, we provide a unified framework - *feature transformers* - for practical lifelong learning to handle data privacy and catastrophic forgetting.

In summary, the major contributions of our work are as follows:

- Define a realistic and a challenging lifelong learning scenario and formulate a unique and generic mathematical framework - feature transformer, to work in this setting successfully.
- Ensure data privacy by storing only features from previous episodes while successfully combating catastrophic forgetting.

- Principled way of utilizing additional neural-compute to tackle complex incremental tasks.

- Demonstrate state-of-the-art results on common continuous learning benchmarks for new task learning like MNIST rotations, iCIFAR dataset and a challenging healthcare problem.

- Demonstrate exemplary performance even under severe constraints on memory and compute, thus operating under the assumptions of practical lifelong learning.

Following review of related work in section 2, we describe the mathematical formulation of feature transformers and their practical realization in section 3. Section 4 contains experiments and results, followed by discussion in section 5.

## 2    RELATED WORK

A conventional deep learning classifier can be viewed as an automatic feature extractor followed by a classifier, trained jointly. For lifelong learning, the methods in literature can be broadly classified into: 1) learning with fixed feature representations and 2) incrementally evolving representations.

**Fixed Represenations**: In this class of approaches, a representation for the data is learnt from the initial task and remains frozen for ensuing tasks, while only the classification layers are modified. The simplest baseline approaches include fine-tuning (Oquab et al., 2014) and Mensink et al. (2012) where the idea of using fixed representations was extended further by using the nearest class mean classifier. In spite of their simplicity, these algorithms do not perform well in practice due to the limiting constraint of a fixed representation throughout the incremental learning phase.

**Incrementally evolving representations**: The last few years have witnessed renewed interest in developing methods that allow changing data representation with the addition of new tasks. Naive methods of retraining with only new data suffer from *catastrophic forgetting* (McCloskey & Cohen, 1989). To overcome this problem, most methods have attempted different manifestations of *rehearsal* (Robins, 1995) - replaying data from the previous tasks. However, access to all the previous data is usually not feasible due to size, compute and privacy concerns, researchers have attempted to reproduce past through proxy information, known as *pseudorehearsal*. Pseudorehearsal techniques can be further classified depending on how the previous information is stored /used:

- **Knowledge distillation based approaches**: An example method for lifelong learning using knowledge distillation (Hinton et al., 2015) is learning without forgetting (LwF) (Li & Hoiem, 2017) - where distillation loss was added to match output of the updated network to that of the old network on the old task output variables. LwF and its extensions does not scale to a large number of new tasks, suffers from catastrophic forgetting and importantly do not address incremental learning or domain adapatation settings.

- **Rehearsal using exemplar sets**: Rebuffi et al. (2017) propose iCaRL, an incremental learning approach by storing an exemplar set of data from the previous tasks and augmenting it with the new data. In more recent works, Javed & Paracha (2018) and Castro et al. (2018) have argued that a decoupled nearest mean classifier from Rebuffi et al. (2017) is not essential and have proposed joint learning of feature extraction and classification. Lopez-Paz et al. (2017) propose Gradient Episodic Memory, a technique which stores data from previous classes and constrains the gradient update while learning new tasks.

- **Rehearsal using generative models**: Triki Rannen et al. (2017) propose to reproduce past knowledge by using task-specific under-complete autoencoders. When a new task is presented, optimization is constrained by preventing the reconstructions of these autoencoders from changing, thereby ensuring that the features necessary for the previous tasks are not destroyed. Shin et al. (2017); He et al. (2018); Venkatesan et al. (2017) employ generative adversarial networks for recreating the history from previous tasks.

- **Network regularization strategies**: Methods belonging to this class aim to identify and selectively modify parts of the neural network which are critical to remember past knowledge or by explicitly penalizing loss of performance on old tasks. Kirkpatrick et al. (2017); Liu et al. (2018) use the Fischer information matrix to identify the most crucial weights responsible for prediction of a given task and lower the learning rate for these tasks. Lee et al. (2017) propose a dynamically expanding network to increase the capacity for new tasks if the previous architecture is insufficient to represent the data.

Our framework lies at the intersection of pseudorehearsal methods and progressive neural networks, with scope for judiciously utilizing extra capacity, while resisting catastrophic forgetting.

# 3 LIFE-LONG LEARNING VIA FEATURE TRANSFORMATIONS

Before we present the feature transformer method, we introduce the terminologies and notations. We consider training a deep neural network which classifies an input to one of the classes $c \in [C] \triangleq \{1, 2, \cdots, C\}$. We refer to the operation of classifying an input to a particular class as a *task*. To this end, the classifier is trained with training dataset $(X, Y)$, drawn from a joint distribution $(\mathcal{X}, \mathcal{Y})$.

We view the deep neural network, defined by the parameters $(\boldsymbol{\theta}, \boldsymbol{\kappa})$, as the composition of a feature extractor $\boldsymbol{\Phi_\theta} : \boldsymbol{X} \to \boldsymbol{F}$, and a classifier $\boldsymbol{\Psi_\kappa}$

$$\boldsymbol{\Psi_\kappa} \circ \boldsymbol{\Phi_\theta} : \boldsymbol{X} \to [C], \tag{1}$$

where $\boldsymbol{X}$ is the space of input data, and $\boldsymbol{F}$ is a space of low-dimensional feature vectors.

We concisely denote the training of the neural network by $\text{TRAIN}(\boldsymbol{\theta}, \boldsymbol{\kappa};\ D)$, which minimizes a loss function on training data $D = (X, Y)$ and produces the network parameters $(\boldsymbol{\theta}, \boldsymbol{\kappa})$. Let us also define the set of all computed features on input $\mathcal{F} = \boldsymbol{\Phi_\theta}(X)$. [1]

When the loss function only penalizes misclassification, the network is expected to learn only the class separation boundaries in the feature space. However, as we demonstrate experimentally, good separation of class specific features enables stable learning of representations, which directly has a bearing on the performance of life-long learning. Therefore, in all our training procedures, we also use a feature loss which promotes feature separation

$$\text{model loss} = \text{classification loss}_{(\boldsymbol{\theta}, \boldsymbol{\kappa})} + \lambda \cdot \text{feature-loss}_{(\boldsymbol{\theta})}. \tag{2}$$

In the lifelong learning context, we denote the time varying aspect of the network, training data and the classes, by using the time symbol $t \in \mathbb{N} \triangleq \{0, 1, 2, \cdots\}$ in superscript on these objects. Realistically, at any time $t > 0$, the classifier encounters any of the following canonical situations:

1. The number of classes remain the same for the classifier as at $t - 1$. The model encounters new training data for a subset of classes, without change in their distribution: $\forall t > 0,\ C^{(t)} = C^{(t-1)},\ \mathcal{T}_1^{(t)} \subseteq [C^{(t)}]$ and $\forall \tau \in \mathcal{T}_1^{(t)},\ (X_\tau^{(t)}, Y_\tau^{(t)}) \sim (\mathcal{X}_\tau^{(t)}, \mathcal{Y}_\tau^{(t)}) = (\mathcal{X}_\tau^{(t-1)}, \mathcal{Y}_\tau^{(t-1)})$.

2. The number of classes remain the same for the classifier as at $t - 1$. The model encounters new training data from modified input distribution(s), for a subset of classes: $\forall t > 0,\ C^{(t)} = C^{(t-1)},\ \mathcal{T}_2^{(t)} \subseteq [C^{(t)}]$ and $\forall \tau \in \mathcal{T}_2^{(t)},\ (X_\tau^{(t)}, Y_\tau^{(t)}) \sim (\mathcal{X}_\tau^{(t)}, \mathcal{Y}_\tau^{(t)}) \neq (\mathcal{X}_\tau^{(t-1)}, \mathcal{Y}_\tau^{(t-1)})$.

3. The model encounters new class(es) and corresponding new data: $\forall t > 0,\ C^{(t)} > C^{(t-1)},\ \mathcal{T}_3^{(t)} = [C^{(t)}] \backslash [C^{(t-1)}]$ and $\forall \tau \in \mathcal{T}_3^{(t)},\ (X_\tau^{(t)}, Y_\tau^{(t)}) \sim (\mathcal{X}_\tau^{(t)}, \mathcal{Y}_\tau^{(t)})$.

In the most generic scenario, a combination of all the three situations can occur at any time index $t$, with training data available for the classes in the set $\mathcal{T}^{(t)} \triangleq \mathcal{T}_1^{(t)} \cup \mathcal{T}_2^{(t)} \cup \mathcal{T}_3^{(t)}$. However, it is important to note that at every index $t$, the classifier is trained to classify all the $C^{(t)}$ classes.

## 3.1 FEATURE TRANSFORMATION BY AUGMENTING NETWORK CAPACITY

At any time $t - 1$, the classifier is optimized to classify all the classes $[C^{(t-1)}]$ and also the set of features $\mathcal{F}^{(t-1)}$ are well separated according to classes. At $t$, when new training data $D^{(t)} = \cup_{\tau \in \mathcal{T}^{(t)}} (X_\tau^{(t)}, Y_\tau^{(t)})$ is encountered, the features extracted using the previous feature extractor

$$\partial \mathcal{F}^{(t)} = \cup_{\tau \in \mathcal{T}^{(t)}} \left( \boldsymbol{\Phi}_{\boldsymbol{\theta}^{(t-1)}} (X_\tau^{(t)}) \right), \tag{3}$$

are not guaranteed to be optimized for classifying the new data and new classes. In order to achieve good performance on new data and classes, we propose to change the feature representation at time $t$, just before the classification stage. We achieve this by defining a ***feature transformer***

$$\boldsymbol{\Phi}_{\Delta \boldsymbol{\theta}^{(t)}} : \boldsymbol{F}^{(t-1)} \to \boldsymbol{F}^{(t)}, \tag{4}$$

---

[1] Though $\boldsymbol{\Phi_\theta}$ is a mapping which acts on individual vectors, we abuse the notation here by using it with sets.

parameterized by $\Delta\boldsymbol{\theta}^{(t)}$, which maps any feature extracted by $\boldsymbol{\Phi}_{\boldsymbol{\theta}^{(t-1)}}$ to a new representation. The new feature extractor is now given by $\boldsymbol{\Phi}_{\boldsymbol{\theta}^{(t)}} \triangleq \boldsymbol{\Phi}_{\Delta\boldsymbol{\theta}^{(t)}} \circ \boldsymbol{\Phi}_{\boldsymbol{\theta}^{(t-1)}}$, where $\boldsymbol{\theta}^{(t)} \triangleq \boldsymbol{\theta}^{(t-1)} \cup \Delta\boldsymbol{\theta}^{(t)}$. Practically, this is realized by augmenting the capacity of the feature extractor at each time $t$ by using one or more fully connected layers[2]. It is however possible that $\boldsymbol{\Phi}_{\Delta\boldsymbol{\theta}^{(t)}}$ could be simply an identity transform and feature transformers learnt in previous episodes could be adapted for new data. This helps in controlling the growth of network capacity over time and this aspect of our work is described in section 4.5.

The feature transformer is trained, along with a new classifier layer, using the composite loss function of the form in equation 2, by invoking TRAIN($\Delta\boldsymbol{\theta}^{(t)}, \boldsymbol{\kappa}^{(t)}; \; D^{(t)}$), with $D^{(t)} = (\partial\mathcal{F}^{(t)}, Y^{(t)})$[3]. This ensures that the classifier performs well on the new data. However, strikingly, training a feature transformer at $t$ does not involve changing the feature extractor $\boldsymbol{\Phi}_{\boldsymbol{\theta}^{(t-1)}}$ at all, and this helps us in alleviating catastrophic forgetting by efficiently making use of already computed features $\mathcal{F}^{(t-1)}$ through a memory module.

### 3.2 Remembering history via memory

The set of all extracted features $\mathcal{F}^{(t-1)}$ serves as a good abstraction of the model, for all the tasks and data encountered till $t-1$. Therefore, if $\mathcal{F}^{(t-1)}$ is available to the model when it encounters new tasks and data, then the feature transformer at $t$ can take advantage of this knowledge to retain the classification performance on previous tasks and data as well. To this end, we assume the availability of an un-ending memory module $\mathcal{M}$, equipped with READ(), WRITE() and ERASE() procedures, that can store $\mathcal{F}^{(t-1)}$ and can retrieve the same at $t$. In situations where memory is scarce, only a relevant subset of $\mathcal{F}^{(t-1)}$ can be stored and retrieved.

We train the feature transformer at any $t > 0$ by invoking TRAIN($\Delta\boldsymbol{\theta}^{(t)}, \boldsymbol{\kappa}^{(t)}; \; D^{(t)}$), with the combined set of features

$$D^{(t)} = (\partial\mathcal{F}^{(t)} \cup \mathcal{F}^{(t-1)}, \cup_{t' \in [1,2,\cdots,t]} Y^{(t')}), \; \forall t > 0. \tag{5}$$

We then obtain the new set of features

$$\mathcal{F}^{(t)} = \boldsymbol{\Phi}_{\Delta\boldsymbol{\theta}^{(t)}}(\partial\mathcal{F}^{(t)}) \cup \boldsymbol{\Phi}_{\Delta\boldsymbol{\theta}^{(t)}}(\mathcal{F}^{(t-1)}), \tag{6}$$

and replace $\mathcal{F}^{(t-1)}$ in memory by (a subset of) $\mathcal{F}^{(t)}$.

With the assumption of infinite memory and capacity augmentation at every episode, our feature transformers framework is presented in algorithm 1.

### 3.3 Feature Transformers in action

Before we present experimental results to demonstrate the ability of feature transformers for lifelong learning of new tasks, we first show how feature transformers operate when simply new episodes of data are presented. As described in section 3, we use a composite loss comprising of classification loss and feature loss to train the classifier. To promote feature clustering/separation, we propose to use *center-loss* as described in Wen et al. (2016).

Dropping the time index $t$ for the brevity, with the convention that the ground truth class for each $x$ is encoded using one-hot vector $\bar{y} = [y_1, y_2, \cdots, y_C]^T$, and $(\boldsymbol{\Psi}_{\boldsymbol{\kappa}} \circ \boldsymbol{\Phi}_{\boldsymbol{\theta}}(x))_c$ is the $c^{th}$ component of the classifier output, we use the following loss function for all the classifier training procedures:

$$\text{loss}(\boldsymbol{\theta}, \boldsymbol{\kappa}) = - \sum_{(x,\bar{y}) \in D} \sum_{c \in [C]} y_c \cdot \log((\boldsymbol{\Psi}_{\boldsymbol{\kappa}} \circ \boldsymbol{\Phi}_{\boldsymbol{\theta}}(x))_c) + \lambda \cdot \sum_{(x,\bar{y}) \in D} \sum_{c \in [C]} \|\boldsymbol{\Phi}_{\boldsymbol{\theta}}(x) - \boldsymbol{\mu}_c\|_2, \tag{7}$$

where $D = \cup_{\tau \in \mathcal{T}}(X_\tau, Y_\tau)$ is the given training data set, and $\boldsymbol{\mu}_c$ is the centroid of all features corresponding to input data labelled as $c$.

Figure 1 provides snapshot of feature transformation algorithm when a new episode of data is encountered. We consider X-Ray lung images (from Wang et al. (2017a)) consisting of two classes: (i) normal and (ii) pneumothorax. At a time index $(t - 1)$, the classifier model is trained on 6000

---

[2]There is no specific restriction on the kind of layers to be used, but in our present work we use only fully connected layers.

[3]$Y^{(t)} = \cup_{\tau \in \mathcal{T}^{(t)}} Y_\tau^{(t)}$

**Input** Task set $\mathcal{T}^{(t)}$, and training data $\cup_{\tau \in \mathcal{T}^{(t)}}(X_\tau^{(t)}, Y_\tau^{(t)}), \forall t \geq 0$
**Output** $(\boldsymbol{\theta}^{(t)}, \boldsymbol{\kappa}^{(t)}), \forall t$

$t \leftarrow 0, \texttt{ERASE}(\mathcal{M})$        /* Set initial time, erase memory */
$D^{(0)} \leftarrow \cup_{\tau \in \mathcal{T}^{(0)}}(X_\tau^{(0)}, Y_\tau^{(0)})$        /* Obtain initial tasks and training data */
$\texttt{TRAIN}(\boldsymbol{\theta}^{(0)}, \boldsymbol{\kappa}^{(0)};\ D^{(0)})$        /* Train initial network */
$\mathcal{F}^{(0)} \leftarrow \cup_{\tau \in \mathcal{T}^{(0)}}(\boldsymbol{\Phi}_{\boldsymbol{\theta}^{(0)}}(X_\tau^{(0)}))$        /* Compute features */
$\texttt{WRITE}(\mathcal{M}, (\mathcal{F}^{(0)}, Y^{(0)}))$        /* Write features to memory */

**while** $TRUE$ **do**
     $t \leftarrow t+1$, obtain $\mathcal{T}^{(t)}, \cup_{\tau \in \mathcal{T}^{(t)}}(X_\tau^{(t)}, Y_\tau^{(t)})$        /* Obtain current tasks and training data */
     Compute $\partial\mathcal{F}^{(t)}$ using equation 3        /* Compute old model features on new data */
     $(\mathcal{F}^{(t-1)}, Y^{(t-1)}) \leftarrow \texttt{READ}(\mathcal{M})$        /* Read previously computed features from memory */
     Form $D^{(t)}$ using equation 5        /* Form composite training data */
     $\texttt{TRAIN}(\Delta\boldsymbol{\theta}^{(t)}, \boldsymbol{\kappa}^{(t)};\ D^{(t)})$        /* Train feature transformer */
     $\boldsymbol{\Phi}_{\boldsymbol{\theta}^{(t)}} \leftarrow \boldsymbol{\Phi}_{\Delta\boldsymbol{\theta}^{(t)}} \circ \boldsymbol{\Phi}_{\boldsymbol{\theta}^{(t-1)}}$        /* Obtain new feature extractor */
     Compute $\mathcal{F}^{(t)}$ using equation 6        /* Compute new features */
     $\texttt{ERASE}(\mathcal{M}), \texttt{WRITE}(\mathcal{M}, (\mathcal{F}^{(t)}, \cup_{t' \in [1,2,\cdots,t]} Y^{(t')}))$        /* Erase & write new features*/
**end**

**Algorithm 1:** The life-long learning framework

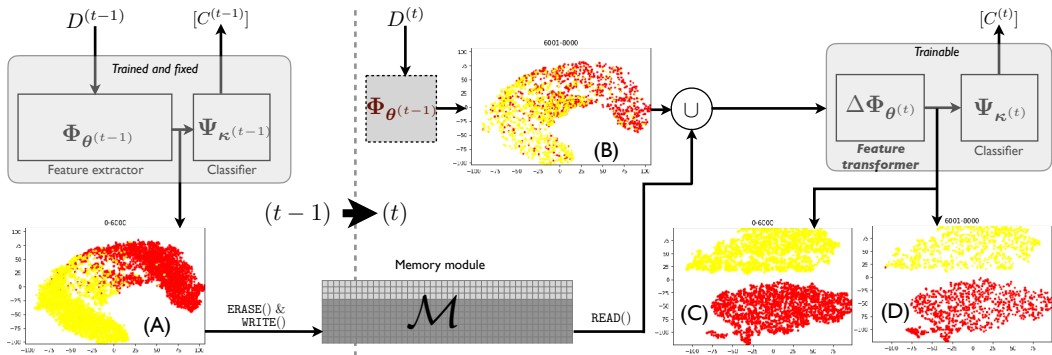

Figure 1: Visual depiction of feature transformation process on new episodes.

images with the loss in equation 7. As shown by the t-SNE plot (A), the feature extractor $\boldsymbol{\Phi}_{\boldsymbol{\theta}^{(t-1)}}$ produces features which are well-separated, and these features get stored in memory $\mathcal{M}$. However, at time $t$, when a set of 2000 new images is encountered, $\boldsymbol{\Phi}_{\boldsymbol{\theta}^{(t-1)}}$ produces features that are scattered (t-SNE plot (B)). To improve the separation on the new data, the feature transformer is trained using the (well-separated) features in $\mathcal{M}$ as well as poorly separated features (from new data), with the loss function promoting good separation in new representation. This ensures that all the 8000 images seen until time $t$ is well separated (t-SNE plots (C) and (D)). This is repeated for all time indices $t$. Thus, the feature transformer, along with appropriate loss function, continuously changes the representation to ensure good classification performance.

## 4 EXPERIMENTAL RESULTS

In this section, we benchmark our algorithm's performance in various scenarios on relevant datasets.

- Realistic Lifelong scenario along with traditional multi-task(MT) on MNIST rotations
- Incremental learning on Pneumothorax identification (data-sensitive domain)
- New-task learning on iCIFAR100

### 4.1 PRACTICAL LIFELONG LEARNER - MNIST ROTATIONS DATASET

To simulate a realistic lifelong learning scenario as described in Section 2, we use MNIST rotations dataset - each task contains digits rotated by a fixed angle between 0 and 180 degrees. Firstly, we

randomly permute the rotation angles (to test domain adaptation). For every rotation angle, we randomly permute the class labels (0-10) and divide them into two subsets of 5 classes each (learning new tasks). We divide the number of training samples into two different subsets (incremental learning). Table 1 details the episodes of a lifelong learning experiment from the described procedure.

| Episode No | Rotation Angle | Class Labels | Samples seen so far/ Total Number of samples available | Description |
|---|---|---|---|---|
| 1 | $\angle(5)$ | $[0, 2, 3, 9, 7]$ | $12.5k/50k$ | Start |
| 2 | $\angle(5)$ | $[6, 4, 5, 1, 8]$ | $25k/50k$ | New-task Learning |
| 3 | $\angle(5)$ | $[0, 2, 3, 9, 7]$ | $37.5.5k/50k$ | Incremental Learning |
| 4 | $\angle(5)$ | $[6, 4, 5, 1, 8]$ | $50k/50k$ | Incremental Learning |
| 5 | $\angle(270)$ | $[2, 8, 6, 4, 7]$ | $62.5k/100k$ | Domain Adaptation |
| 6 | $\angle(270)$ | $[5, 0, 1, 3, 9]$ | $75k/100k$ | Domain Adaptation |
| 7 | $\angle(270)$ | $[2, 8, 6, 4, 7]$ | $87.5k/100k$ | Domain Adaptation + Incremental Learning |
| 8 | $\angle(270)$ | $[5, 0, 1, 3, 9]$ | $100k/100k$ | Domain Adaptation + Incremental Learning |
| .. | .. | .. | .. | .. |
| .. | .. | .. | .. | .. |

Table 1: Episodes descriptions in one realistic life-long learning experiment scenario

### 4.1.1 EXPERIMENT DETAILS AND RESULTS

We used a basic CNN architecture composed of 3 conv-layers of 32 filters and kernel-size - (3x3) followed by 2 dense layers "fc1" and "fc2" of feature length 256 and a softmax. Our feature transformer networks at every episode aims to transform "fc1" features using 2 additional dense layers of feature length 256. Feature transformers from previous episode serve as initialization for current episode and these models were optimized for the cumulative loss (equation 7), with $\lambda = 0.2$. All the feature transformers were trained for only 3 epochs, with batch size of 32.

We compare our results to the two obvious life-long learners - naive and cumulative training, which serve as the lower and upper bounds of performance respectively in the absence of other competitive algorithms for this setting. While the naive learner finetunes the entire network on the latest episode data, cumulative learner accumulates data from all the episodes seen so far and retrains the model. Fig. 2a highlights the remarkable performance of the proposed approach. We show the performance evolution over the first 25 episodes in Fig. 2a while, performance comparisons over 80 episodes across multiple experiment runs are averaged and shown in Table. 2b. We demonstrate that just by storing features in memory and learning feature transformers at every episode, we achieve almost similar performance to the gold-standard result. Further, it also underlines the applicability of our approach as single framework to combat different requirements of lifelong learning.

Finally, we also compare our results (figure 3) in the conventional MT setting described in Lopez-Paz et al. (2017), whose open source implementation we used for our experiments. For details on the terminology and compared methods, the readers are encouraged to refer to Lopez-Paz et al. (2017). Fig. 3a demonstrates clear superiority of our approach over multiple methods compared. Additionally, *backward transfer* (BWT) - quantitative metric that models the deterioration of performance on older tasks while learning new tasks, is also negligible for our method, which shows resistance to catastrophic forgetting. Fig. 3b highlights this further as shown by accuracy on first task after learning subsequent tasks.

### 4.2 PNEUMOTHORAX IDENTIFICATION FROM X-RAY IMAGES

We simulate a practical manifestation of lifelong learning where a model trained to detect pneumothorax is deployed in a hospital with data arriving incrementally. We utilize a subset of ChestXRay (Wang et al., 2017b) dataset, which consists of chest X-rays labeled with corresponding diagnoses. We simulated incremental learning by providing the 8k training images in incremental batches of 2k and measured the performance on held-out validation set of 2k images. Fig. 4a establishes the baselines for the experiment. As in previous experiment, naive and cumulative training define the performance bounds. To clearly highlight the value of our feature transfom, we also add another strong baseline - naive Learner with center loss, which learns on the recent batch but with an augmented loss function (equation 7). In spite of a gain of 5% due to center loss, there is still a loss of 4% performance in the incremental learning set-up in the fourth batch of 2k.

**Experimental Details and Results**: We used a pre-trained VGG-network (Simonyan & Zisserman, 2014) as the base network and explored the use of features from different layers of the VGG-network

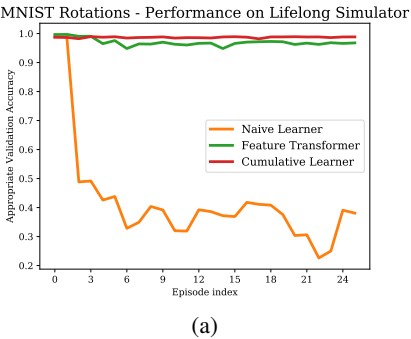

Figure 2: (a) Performance evolution over first 25 episodes and (b) Average performance across multiple runs at the end of lifelong experiments

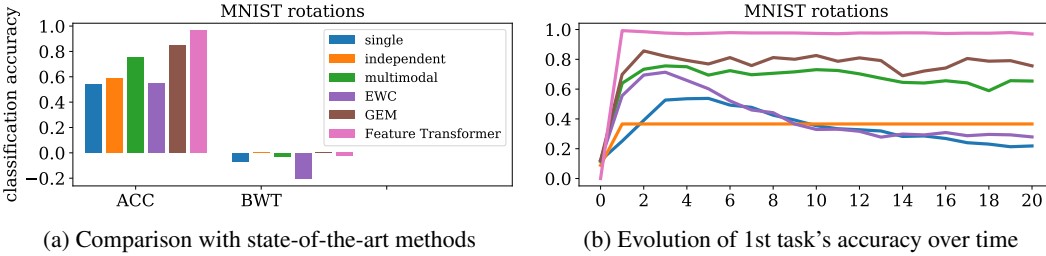

(a) Comparison with state-of-the-art methods     (b) Evolution of 1st task's accuracy over time

Figure 3: Comparison of proposed approach in conventional multi-task setting

namely - post the two pooling layers and fully connected layers. Feature transformer network essentially had one additional dense layer per step and was optimized for (equation 7).

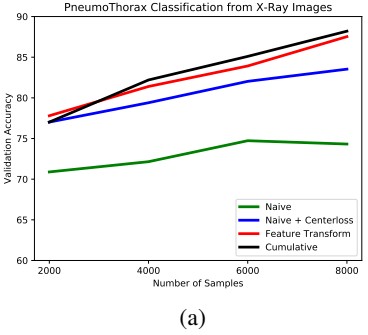

Figure 4: (a) Performance Comparison on validation dataset and (b) Comparison of feature transformers from different base layers

Fig. 4a captures the performance of feature transformer with the base features being extracted from first pooling layer - block3_pool. After fourth batch of data, feature transformer result almost matches the performance of cumulative training. This performance is achieved despite not having access to the full images but only the stored features. Table. 4b also presents the performance of feature transformer depending upon the base features used. It can be noted that performance is lowest for the layer that is closer to the classification layer - fc_2. This is intuitively satisfying because, the further layers in a deep neural network will be more finely tuned towards the specific task and deprives the feature transform of any general features.

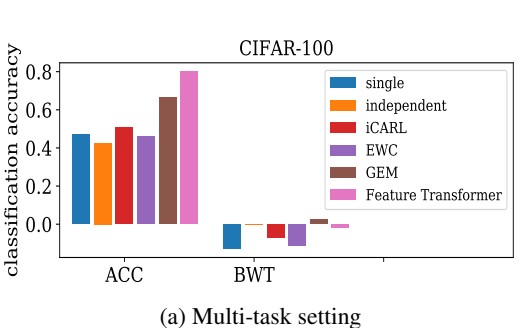

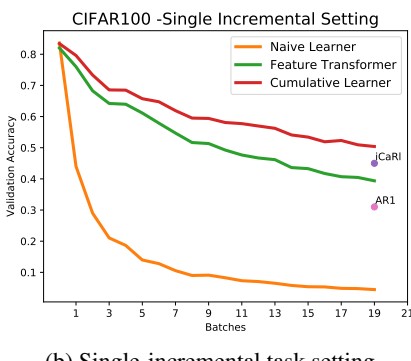

(a) Multi-task setting

(b) Single-incremental task setting

Figure 5: Comparison with state-of-the-art methods - iCIFAR100 dataset

### 4.3 ICIFAR100 DATASET

We present the 100 classes from CIFAR100 dataset in a sequence of 20 tasks comprising of 5 classes each. Similar to Lomonaco & Maltoni (2017), we use the definitions for MT and single-incremental task (SIT). In an MT setting, evaluation is performed only on the new tasks exposed to the learner in the current episode. We start with a VGG type architecture, pretrained on iCIFAR10 dataset as suggested by Lopez-Paz et al. (2017) and Lomonaco & Maltoni (2017). Base features are extracted from flatten layer (before the fully connected layers) and our feature transformers included two dense layers with feature length = 256. Similar to our earlier experiments, the feature transformer module from the previous episode initializes the current episode transformer and is optimized for the cumulative loss, with NoEpochs = 30 and batch_size = 32.

#### 4.3.1 MULTI-TASK SETTING

Fig 5a demonstrates the superiority of our approach by a significant margin of $>10\%$. Further, we have negligible Backward Transfer.

#### 4.3.2 SINGLE INCREMENTAL TASK SETTING

Fig 5b captures the performance over 20 batches of data for our method along with cumulative and naive learner. Unsurprisingly, naive learner performs very poorly, while feature transformer displays exemplary performance numbers of $40\%$ validation acccuracy after encountering all 20 episodes, while compared to cumulative learner at $50\%$. Our method significantly outperforms AR1 (Lomonaco & Maltoni, 2017), while iCaRL (Rebuffi et al., 2017) achieves best-in-class performance close to gold-standard cumulative learner. This is not surprising because iCaRL is an explicit rehearsal technique, where exemplar images from previous episodes are stored and replayed while learning new tasks, while we only store low dimensional features.

### 4.4 EFFECT OF LIMITED MEMORY

One of the drawbacks of the proposed approach is the assumption of infinite memory and need to store features computed on all samples observed so far. To understand the extent of this limitation, we performed ablation experiments limiting the amount of history replayed as well as computing the storage requirements involved.

***Storing all features is not necessary*** We studied the effect of size of memory by limiting the number of samples stored to a smaller percentage. We observed that performance dropped from $97\%$ to $94\%$ when we reduced the memory size to only 25% the original size on MNIST rotations (Table. 2). We performed similar experiments on Pneumothorax classification problem and achieved similar trends as shown Table. 3, clearly demonstrating the resistance of the proposed method.

***Storing all features is not prohibitive*** Additionally, calculations of size-on-disk suggests that storing features of the entire history is not prohibitive. A typical natural/medical image is 256*256*3 integers or more, whereas our representation is only 4096 floats (16kb). Even the largest available medical image repository of 100k X-ray images takes 1.6GB which is not huge. These are con-

| % of Samples stored from history | Feature Transformer Val Acc | Cumulative Learner Val Acc |
|---|---|---|
| 25 | 94% | 95% |
| 50 | 94.8% | 98% |
| 75 | 96% | 98.5% |
| 100 | 97 % | 99 % |

Table 2: MNIST rotations

| % of Samples stored from history | Feature Transformer Val Acc |
|---|---|
| 25 | 80.95% |
| 50 | 82.95% |
| 75 | 85.85 % |
| 100 | 86.94 % |

Table 3: Pneumothorax dataset

Table: Performance comparisons with limited memory budget

servative estimates. A standard medical image can be of much larger size (1024*1024) and in 3-D (minimum 10 slices). Any exemplar-based method (iCARL) will have severe storage limitations than our method. Additionally, storing 50 low-dimensional features occupies same memory as storing one exemplar image. This directly leads to storing more history compactly while addressing catastrophic forgetting and privacy.

### 4.5 CONTROLLING THE GROWTH OF NETWORK CAPACITY

In the description of the feature transformer framework in section 3.1 and section 3.2, we provided a generic treatment of the method, where, at the end of the each episode, the features are transformed up to date and then stored in memory for the next episode. With this scheme, it becomes imperative to always augment the capacity of the network in order to learn new representations, resulting in ever-growing network capacity. However, this problem can be easily alleviated by partitioning the entire network into a base feature extractor and feature transformer layers. The base feature extractor remains always fixed, and it is only the output of base feature extractor that is always stored in memory. With this scheme, the feature transformer layers need not grow in capacity and only a few already existing layers can be adapted for the new data. When the capacity of the feature transformer layers is not sufficient, then it can be augmented by adding one or more extra layers. In either case, the stored base features are sufficient to train.

***Effect of varying additional capacity***

We varied the size of feature transformers and observed the difference in performance. Table 4 shows that by halving the additional capacity does not change the performance on MNIST rotations dataset at all. In addition, we froze the capacity of feature transformers after 5th episode and adapted them till end of 80 episodes. It is striking that performance is still high. Similarly, for Pneumothorax classification, Table 5 shows the performance comparisons with varying capacity of 2 , 1 and zero fc layers post third episode. These experiments (along with Sec 4.4) clearly demonstrate that power of the proposed approach comes from learning separable representations continually and not necessarily from storing all features or additional capacity.

| Incremental capacity added per episode | Feature Transformer Val Acc |
|---|---|
| 2 dense layers | 96.4% |
| 1 dense layers | 96.5% |
| No additonal capacity (after $5^{th}$ episode) | 96.2% |

Table 4: MNIST rotations

| Incremental capacity added per episode | Feature Transformer Val Acc |
|---|---|
| 2 dense layers | 86.94% |
| 1 dense layers | 86.43% |
| No additonal capacity (after $3^{rd}$ episode) | 86.41% |

Table 5: Pneumothorax dataset

Table: Performance comparisons with limited incremental compute

## 5 DISCUSSION

In the final section, we discuss various points concerning our proposed approach.

***Bayesian interpretation of Feature transformers***

Our feature transformers framework - learning a new representation with every new episode of data, can be interpreted as a *maximum-a-posteriori* representation learning, with the previous representation acting as prior. The implementation described in this paper using the combination loss (equation 7) is one instantiation of a general incremental representation learning possible in our framework. We have cast the MAP estimate problem to a tractable optimization problem constrained by a center-loss. In future, we plan to explore other manifestations of our approach with different loss functions that can ensure better *separability*.

### Information Loss, Incremental Capacity, Data Privacy

As shown in Table. 4b, feature transformer becomes less effective if the base features do not contain enough relevant information. This also means that additional capacity that every feature transformer adds may not help or in-fact be counter-productive. If the base features are extracted from layers close to the input image, there will be problem of traceability which violates the data privacy requirement we want to accomplish. We feel this a potential trade-off between performance and data privacy which we will investigate in future.

### Model compaction for cascade of feature transformers

Another approach to control the growth of network capacity is model compaction, which will be our future work. At any point in time, the entire set of feature transformer layers can be replaced by a smaller and simpler network (possibly using distillation techniques) and then again allowed to grow subsequently. This cycle of grow-and-purge can be used to effectively manage the overall capacity of the network.

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
