# OpenReview forum: "Feature Transformers: A Unified Representation Learning Framework for Lifelong Learning"
_ICLR.cc/2019/Conference_

### Official Review · AnonReviewer3 · 2018-11-01
**Mechanical Approach to Augmenting Networks Incrementally for Lifelong Learning**

**Rating:** 4
**Confidence:** 5

**Review:**

The authors provided a training scheme that ensures network retains old performance as new data sets are encountered (e.g. (a) same class no drift, (b) same class with drift, (c) new class added). They do this by incrementally adding FC layers to the network, memory component that stores previous precomputed features, and the objective is a coupling between classification loss on lower level features and a feature-loss on retaining properties of older distributions. The results aren't very compelling and the approach looks like a good engineering solution without strong theoretical support or grounding.

---

> ### Author Response · Authors · 2018-11-28
> **State of the art results**
>
> We would like to thank the reviewer for their time and patience.
>
> The reviewer has made 2 important remarks on the paper:
> 1. Non compelling results
> 2. An engineering solution
>
> 1. Non compelling results
> To the best of our knowledge, we have achieved state-of-the-art results on multiple variants of continual learning. In fact, we would like to take this opportunity to point out that our method outperforms all algorithms on the different variants of continual learning (new task learning(single incremental task/multi-task, incremental learning, domain adaptation). We would like the reviewer to point us to other works which can help us benchmark our results better.
>
> 2. An engineering solution
> Continual learning is an important problem with a large number of practical use-cases in the industry across various domains. It is important that such a solution is not only of academic interest but also can be deployed practically. We believe that an innovative solution, which does not compromise privacy of historical data while enabling continual learning will be of sufficient interest to the community and will spur further research in this direction.

---

### Official Review · AnonReviewer1 · 2018-11-02
**The work present a framework for dealing with life long learning, yet it violates two important constraints which every life long learner has to obey: limited memory and computation.**

**Rating:** 3
**Confidence:** 4

**Review:**

Summary:
 a method is presented for on-going adaptation to changes in data, task or domain distribution. The method is based on adding, at each timed step, an additional network module transforming the features from the previous to the new representation. Training for the new task/data at time t relies on re-training with all previous data, stored as intermediate features. The method is shown to provide better accuracy than naïve fine tuning, and slightly inferior to plain re-training with all the data.
While the method is presented as a solution for life long learning, I think it severely violates at least two demands from a feasible solution: using finite memory and using finite computational capacity (i.e. a life-long learning cannot let memory or computation demands to rise linearly with time). Contrary to this, the method presented induces networks which grow linearly in time (in number of layers, hence computation requirements and inference time), and which use a training set growing indefinitely, keeping (representations of) all the examples ever seen so far. If no restrictions on memory and computation time are given, full retraining can be employed, and indeed it provides better results that the suggested method. In the bottom line, I hence do not see the justification for using this method, either as a life-long learner or in another setting.

Pros:
+ the method shows that for continuous adaptation certain representations can be kept instead of the original examples
Cons:
- The method claims to present a life long learning strategy, yet it is not scalable to long time horizon (memory and inference costs rise linearly with time)
- Some experiments are not presented well enough to be understood.

More detailed comments:
Page 3:
-	Eq. 2 is not clear. It contains a term ‘classification loss’ and ‘feature_loss’ which are not defined or explained. While the former is fairly standard, leaving the latter without definition makes this equation incomprehensible.
o	I later see that eq. 7 includes the details. Therefore eq.2 is redundant.
Page 4:
-	Eq. 5 seems to be flawed, though I think I can understand what it wants to say. Specifically, it states two sets: one of examples (represented by the previous feature extractor) and one of labels (of all the examples seen so far). The two sets are stated without correspondence between examples and labels – which is useless for learning (which requires example-label correspondence). I think the intention was for a set of (example, label) pairs, where the examples are represented using feature extractor of time t-1.
-	Algorithm 1 seems to be a brute force approach in which the features of all examples from all problems encountered so far are kept (with their corresponding labels). This means keeping an ever growing set of examples, and training repeatedly at each iteration on this set. These are not realistic assumptions for a life-long learner with finite capacity of memory and computation.
o	For example, for the experiment reported at page 6, including 25 episodes on MNist, each feature transformer is adding 2 additional FC layers to the network. This leads to a network with >50 FC layers at time step 25 – not a reasonable and scalable network for life ling learning
Page 6:
-	The results show that the feature transformer method achieve accuracy close to cumulative re-training, but this is not too surprising, since feature transformer indeed does cumulative re-training: at each time step, it re-trains the classifier (a 2 stage MLP) using all the data at all times steps (i.e. cumulative retraining). The difference from pure cumulative re-training, if I understand correctly, is that the cumulative re-training is done not with the original image representations, but with the intermediate features of time t-1. What do we earn and what do we loose from this? If I understand correctly, we earn that the re-training is faster since only a 2-layer MLP is re-trained instead of the full network. We loose in the respect that the model gorws larger with time, and hence inference becomes prohibitively costly (as the network grows deeper by two layers each time step). Again, I do not think this is a practical or conceptual solution for life long learning.
-	The experiment reported in figure 3 is not understandable without reading Lopez-Paz et al., 2017 (which I didn’t). the experiment setting, the task, the performance measurements – all these are not explained, leaving this result meaningless for a stand-alone read of this paper.
-	Page 8: it is stated that “we only store low dimensional features”. However, it is not reported in all experiment exactly what is the dimension of the features stored and if they are of considerably lower dimension than the original images. Specifically for the MNIst experiments it seems that feature stored are of dimension 256, while the original image is of dimension 784 – this is lower, but no by an order of magnitude (X10).
-	The paper is longer than 8 pages.

---

> ### Author Response · Authors · 2018-11-28
> **Our method uses practically feasible memory and compute**
>
> Thank you, reviewer for your detailed comments on the paper.  From the insight gained from reviews, we have accordingly modified the paper, along with a note of changes in our  comment, "Major points revisited / concerns addressed" on this forum.
>
> As a couple of reviewers had similar concerns,  we have addressed  these through a general comment about managing increasing memory and compute. We argue that our method's memory and compute requirement, owing to storage of only a fraction of historical lower dimensional features and requiring the use of additional compute only optional, is practically feasible and of value to the community.
>
> Some of the specific concerns which are addressed are:
>
> i. Redundancy of equation 2 by 7.
> We have structured our paper to first paper to first present a broad overview without being cluttered with design choices. Our paper's design was motivated to ensure we first convey the broad idea followed by a specific embodiment which was implemented.
>
> ii. Errors in Equation 5
> As the reviewer pointed out, this equation represents the construction of training data for the feature transformer at time t, in the form of example-label pairs. Though not explicitly mentioned, it should be understood from the context that indeed the correspondence between examples (in the form of features) and labels are maintained due to the order of the union operation. If it adds value to make this point explicit, we would be happy to include this clarification in the next version of the paper.
>
> iii. Increasing memory/compute
> Both these concerns are addressed through the general comment, "Major points revisited / concerns addressed".
>
> iv. Add experimental details to ensure paper is stand-alone
> We have incorporated the comments by reviewers to ensure that all definitions and experimental settings  are fully contained in the paper and can be understood unambiguously.
>
> v. Memory gains by storing features
> While we agree with the reviewers that memory gains by storing features are limited when working with a toy dataset like MNIST(28x28), as pointed out in the general comment, these gains are substantial when working with real-world datasets, and become extremely pronounced in volumetric datasets like medical imaging (3D Datasets).
>
> vi. Paper longer than 8 pages
> We consciously wanted to ensure verbosity so that the subtle ideas are conveyed meaningfully.

---

### Official Review · AnonReviewer2 · 2018-11-02
**Continual learning approach with increasing computational cost over time**

**Rating:** 4
**Confidence:** 3

**Review:**

This paper proposes a continual learning approach which transforms intermediate representations of new data obtained by a previously trained model into new intermediate representations that are suitable for a task of interest.
When a new task and/or data following a different distribution arrives, the proposed method creates a new transformation layer, which means that the model’s capacity grows proportional to the number of tasks or data sets being addressed over time. Intermediate data representations are stored in memory and its size also grows.
The authors have demonstrated that the proposed method is robust to catastrophic forgetting and it is attributed to the feature transformation component. However, I’m not convinced by the experimental results because the proposed method accesses all data in the past stored in memory that keeps increasing infinitely. The authors discuss very briefly in Section 5.2 on the performance degradation when the memory size is restricted. In my opinion, the authors should discuss this limitation more clearly on experimental results with various memory sizes.

The proposed approach would make sense and benefit from storing lower dimensional representations of image data even though it learns from the entire data over and over again.
But it is unsure the authors are able to claim the same argument on a different type of data such as text and graph.

---

> ### Author Response · Authors · 2018-11-28
> **Response to AnonReviewer2: Additional computational cost is limited**
>
> Thank you for your insights on the paper.  Taking your comments into consideration, we have accordingly modified the paper, along with a note of changes in our general comment, "Major points revisited / concerns addressed" on this forum.
>
> The major concerns raised are:
> i. increasing computational cost (number of layers) with time
> ii. increasing memory requirement with time
> iii. Experiments on different types of datasets (text/graph).
>
> Points (i and ii): The first couple of points are addressed in our general comment, where we argue that these requirements are not limiting and will not constrain the implementation of a lifelong learner.
>
> Point iii:  The focus of this work was on developing a secure, privacy-aware continuous learning system for domains involving images (e.g., medical imaging). While we haven't performed any experiments on text/graphs, we believe that the method is generic enough for application in other forms of data. We plan to validate this method on other data types in the future.

---

### Author Response · Authors · 2018-11-27
**Major points revisited / concerns addressed**

Thanks to all the reviewers for their time and patience. Before we make rebuttals to specific concerns, we highlight a few important facets of the work, which may have been overlooked.

1) Privacy-preserving lifelong learners : A key constraint that we operate is that we cannot store data in original form. This is a very pertinent point in data-sensitive domains like Healthcare, where images of subjects cannot be shipped/stored for anonymity/regulatory constraints, which is a point often overlooked. The fundamental question that we address is : "Can we protect privacy while also ensuring competent lifelong learning?" To that end, we demonstrate that, by merely storing low-dimensional features, we can achieve a practical lifelong learner without compromising security.

2) Single-framework for lifelong learning:  The work addresses all variants of continuous learning together, which has not been addressed yet in the current literature to the best our knowledge. E.g. a method that addresses new-task learning cannot handle incremental learning of same task (LwF) or a method that achieves domain adaptation  may not handle new-task learning or incremental learning. Our framework is capable of handling a variety of situations without retraining the entire network, and we have shown the ability of our method to handle real-life type situations, where we simulate complex sequences of these situations.

3) Significance of results: The reviewers have not appreciated the exemplary results on three different datasets. We would like to point out that even with the luxury of having all the low dimensional features from previous episodes, obtaining state-of-the art results is non-trivial. This is achieved by seeking newer representations that continue to be separable, and cannot be attributed only to increasing capacity.

We proceed to address the two major concerns of the reviewers - Memory and Computation

1) Memory: We do not think storing all features is a stumbling block for realizing the potential of our method as discussed here.

	a) Storing all features is not necessary

	As demonstrated in Section 4.4, by storing only one-fifth of past history, we observe a drop in performance of only ~3% from baseline. To our knowledge, such a performance is extremely compelling and difficult to achieve. We have included an additional experiment on Pneumothorax classification, where we achieve significant performance by retaining only 20% of previous data at every episode. We have captured these ablation experiments in graphical format in the revised version along with size-on-disk computation.

	b) Storing all features is not prohibitive

	Additionally, calculations of size-on-disk suggests that storing features of the entire history is not prohibitive. A typical natural/medical image is 256*256*3 integers or more, whereas our representation is only 4096 floats (16kb). Even the largest available medical image repository of 100k X-ray images takes 1.6GB which is not huge. These are conservative estimates. A standard medical image can be of much larger size (1024*1024) and in 3-D (minimum >10 slices). Any exemplar-based method (iCARL) will have severe storage limitations than our method. Additionally, storing ~50 low-dimensional features occupies same memory as storing one exemplar image.  This directly leads to storing more history compactly while addressing catastrophic forgetting and privacy.

2) Computation: We highlight that incremental computational requirements are not unrealistic as suggested.

	a) Additional compute not prohibitive :

		As time progresses, only new layers have to be learnt and it is not entirely true that the training overhead is costly. In our experiments, we have shown that the feature transformer layers are typically two layer deep and hence learning them is not expensive. Further, we have added results where we trained with only one additional fc layer, which did not give any drop in performance. This coupled with storing a fraction of historical data, makes training and inference light.

	b) Additional compute may not be needed :

		The intuition behind adding more capacity is to account for cases where existing representations might be insufficient to achieve separation. There is no compulsion  to add capacity at each episode (Sec 3.1). We have added the experiment (Sec 4.5) where we do not add additional capacity after 5th episode. We simply transform the previous episode's feature transformer to achieve separable representation. However, we wanted to present a generic framework that can account for the need to seek richer representations for complex lifelong problems.

	c) Model Compaction :

		Another idea that has already been mentioned in the paper is model compaction. Distillation based approaches have demonstrated 	shrinking of huge networks without loss of performance, which can be employed in this work.

---

### Meta-Review · Area_Chair1 · 2018-12-13

**Confidence:** 5
**Recommendation:** Reject

**Metareview:**

The paper proposes a framework for continual/lifelong learning that has potential to overcome the problems of catastrophic forgetting and data privacy.
R1, R2 and AC agree that the proposed method is not suitable for lifelong learning in its current state as it linearly increases memory and computational cost over time (for storing features of all points in the past and increasing model capacity with new tasks) without account for budget constraints.

The authors responded in their rebuttal that the data is not stored in the original form, but using feature representation (which is important for privacy issues). The main concern, however, was about the fact that one has to store information about all previous data points which is not feasible in lifelong learning. In the revision the authors have tried to address some of the R1’s and R2’s suggestion about taking into account the budget constraints. However more in-depth analysis is required to assess feasibility and advantage of the proposed approach.
The authors motivate some of the key elements in their model as to protect privacy. However no actual study was conducted to show that this has been achieved.
The comments from R3 were too brief and did not have a substantial impact on the decision.

In conclusion, AC suggests that the authors prepare a major revision addressing suitability of the proposed approach for continual learning under budget constraints and for privacy preservation and resubmit for another round of reviews.